# Physical Computing Strategy to Support Students' Coding Literacy: An Educational Experiment with Arduino Boards

**Chih-Chao Chung and Shi-Jer Lou \*** 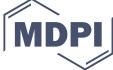

General Research Service Center, Graduate Institute of Technological and Vocational Education,
National Pingtung University of Science and Technology, Pingtung 912, Taiwan; ccchung@g4e.npust.edu.tw
\* Correspondence: lou@mail.npust.edu.tw

**Abstract:** The purpose of this study is to explore the influence of introduction of the physical computing strategy of Arduino Boards in a program design course on coding literacy and the effectiveness of the application in technical high school students. This study selected two classes of twelfth-grade students enrolled in a program design course at a technical high school in Southern Taiwan as the samples. One class was the control group (43 students), and the other was the experimental group (42 students). During the 18-week course, the control group carried out a DBL (design-based learning) programming project, and the experimental group carried out the DBL programming project using the physical computing strategy of Arduino boards. Pre- and posttests and a questionnaire survey were carried out, while ANCOVA (analysis of covariance) was used for evaluation purposes. In the course, students in the experimental group were randomly selected for semi-structured interviews to understand their learning status and to perform qualitative analysis and summarization. This study proposed the physical computing strategy of Arduino boards, featuring staged teaching content, practical teaching activities, and real themes and problem-solving tasks. The results show that the coding literacy of students in the different teaching strategy groups was significantly improved. However, in the Arduino course on DBL programming, the students in the experimental group had a significantly higher learning efficiency in coding literacy than those in the control group. Moreover, according to the qualitative analysis using student interviews, Arduino boards were found to improve students' motivation to learn coding and to aid in systematically guiding students toward improving their coding literacy by combining their learning with DBL theory. Thus, Arduino technology can be effectively used to improve students' programming abilities and their operational thinking in practically applying programming theories.

**Keywords:** DBL; Arduino; coding literacy; physical computing; education reform

## 1. Background

The cultivation of information technology abilities has been an educational goal to which all countries in the world have attached importance. For example, in 2016, the United States proposed the plan of Computer Science for All [1] and invested USD 4 billion to promote programming courses in seven of the largest school districts in the United States, with the aim of motivating students from kindergarten to high school to learn information science and to cultivate an ability to create a digital economy [2]. In addition, the American College Board introduced the AP Computer Science Principles for Advanced High School Computer Science Courses, which focuses on the application of information technology and programming problem-solving [3]. Many advanced developing countries around the world, including 15 countries in Europe, the United Kingdom, Australia, and many others, have officially incorporated programming into their school curriculum. In 2018, Taiwan also included the learning of "computational thinking" and "programming language" into the curriculum of science and technology [4] in the general outline of its 12-year national basic education curriculum. It is obvious that the cultivation of programming abilities has become one of the important indicators of all stages of education.

Many studies have pointed out that learning programming can cultivate high-level problem-solving abilities in students and can help them develop their career in the future [5]. However, programming is a difficult subject to teach and to learn in a technical high school. Traditional programming teaching focuses too much on syntactical teaching, which makes it easy for beginners to fall into the trial-and-error way of piecing together programs [6]. Most students at the beginning of learning programming often encounter difficulties in syntax and program concepts due to the huge and complex syntactical structure of general programming language, resulting in difficulties and frustration for program beginners [7]. Therefore, trying to use different teaching methods or designing diversified course contents to enhance the effectiveness of students' learning of programming is an important topic of educational institutions in various countries.

Design-based learning (DBL), which is often used in the teaching field, presently emphasizes design thinking, hands-on practice, and problem-solving abilities. The repeated thinking, practical activities, and problem-solving process of DBL conform to the teaching objectives of a physical computing programming course [8]. Moreover, in addition to the teaching of Scratch programming, the design of the information course is more integrated into the application of Arduino-related open hardware. From program education originally operated by computer software to the implementation experience imported into hardware, the programming of an information course achieves a real interactive effect through the control of various sensors and acousto-optic components [9,10]. Therefore, the integration of design thinking and physical computing, focusing learning activities on solving complex tasks and producing solutions to future problems [11], as the main axis of teaching and students' learning in this study is expected to achieve good results. In view of this, this study aims to determine the impact of the integration of Arduino, which applies DBL and a physical computing strategy, in a technical high school programming project course on the cultivation of coding literacy in technical high school students.

## 2. Literature Review

### 2.1. Computational Thinking and Coding Literacy for Programs

Professor Jeannette M. Wing argued that computer operation should be added to basic language skills. Besides reading, writing, and arithmetic skills, the concept of computer operation should also be taught: "the skill of computer operation is not only reserved for computer scientists, but an ability and literacy that everyone should have." [12,13]. Computational thinking is the thinking process involved in solving problems, designing systems, and understanding human behavior when performing computer science-related work. Therefore, programming is a cognitive activity that requires logical reasoning abilities. Through the learning process of programming, students' high-level thinking and logical reasoning abilities can be cultivated [14].

In the process of writing programs, students must first understand the problem and figure out the method (algorithm) to solve the problem, which denotes the stage of problem solving. The solution of the problem is translated into code, which is the so-called stage of program writing and practice. In the stage of problem solving, it is very difficult for beginners to learn programming. Linn and Clancy [15] noted that previous programming teaching focused too much on syntax and neglected to teach the importance of problem-solving abilities. How to teach students to apply general problem-solving abilities to programming is an important issue in programming teaching [16]. Valente [17] pointed out that the "description-execution-reflection-debugging-description" cycle that is repeated in the programming process is the most effective programming learning technique and training method for students [18]. Therefore, in addition to strengthening students' logical thinking abilities, programming education is helpful for improving their algorithm skills, knowledge of learning computer concepts, and problem-solving skills [19].

## 2.2. Physical Computing

Physical computing (also known as physical interaction design) involves building interactive physical systems using software and hardware that can perceive and respond to the simulated world. In other words, physical computing is an innovative framework for understanding the relationship between human beings and the digital world, which is not a new technology but rather an integrated system of many technologies. In practical applications, physical computing uses a microcontroller, sensor, and actuator to convert analog inputs into a software system or to control electronic mechanical devices, such as motors, steering gears, lighting equipment, or other hardware. It therefore acts as a bridge between people and people, people and things, and things and things or as interactive platform for handmade works of art, design, or DIY (Do It Yourself) projects [20], including interactive media art works, integrating various novel human–computer interaction methods, such as electronic fish tank, interactive projection, virtual book flipping, 4D cinema, etc. [21]. It is mainly formed by connecting an interactive device system, constructed by a microcontroller, sensor, and actuator, with the real world and strengthening the seamless connection between the digital virtual world and the real simulated world (also known as the combination of virtual and real). The working principle behind these interactive media works is physical computing.

Interactive media works created based on the principle of physical computing include three key links: motion acquisition (diversified input), intelligent processing (rapid and accurate processing), and display output (diversified output). In order to realize these key links, a variety of technologies must be integrated, such as sensor technology, control technology, programming technology, network communication technology, and so on. For example, robotics, an integrated curriculum, and technology were integrated by MIT (Massachusetts Institute of Technology) in the development of computer educators in the 1990s as tangible programming bricks that could be combined with sensors to make programming bricks interact with the environment, which later became Lego Mindstorms [22].

The Arduino development tool, which is the main architecture of today's Internet of Things (IoT), is also the mainstream implementation course. Arduino is an open-source single-chip microcontroller, which has both a software and hardware architecture. It was jointly developed by the Italian teachers, Massimo Banzi et al., for the purpose of teaching. The current architecture of IoT also uses the Arduino core as a development tool, providing a good auxiliary tool for users who are interested in creation. Arduino integrates a development environment that is characterized by (1) a low price, (2) development across various operating system platforms, and (3) a simple integrated program development environment, in which the software and hardware architecture can easily be expanded and made open-source [23].

The Arduino programming language is a skill-learning software and hardware collocation. The hardware part needs to be equipped with instruments, tools, and wire components. Arduino can be expanded to have multiple input and output devices and sensors, including sliders, buttons, buzzers, temperature sensors, humidity sensors, LED modules, photosensitive resistance modules, smoke sensors, alcohol sensors, etc. Therefore, Arduino is also the mainstream implementation course introduced into engineering-related courses at present. The research results show that students' learning motivation and learning effectiveness can be effectively improved through Arduino learning programming [24]. Arduino teaching aids are often used in technical high school programming courses. Especially for student beginners, the characteristics of interaction and real-time feedback can attract students' interest, arouse their learning motivation, and thus improve their computational thinking abilities [9]. Some researchers have pointed out that the application of Arduinos in programming courses can help improve the effect on students' learning without an information background in the course [25] and can help to strengthen teacher–student interaction and to improve the effect on students' learning [26,27].

This study therefore mainly discusses the impact of DBL combined with the open-hardware physical computing of Arduino on students' coding literacy in a technical high school programming course as well as the possible problems encountered in the course and countermeasures to provide a reference for information technology teachers and future researchers.

*2.3. Design-Based Learning*

Design-based learning is an exploratory learning activity that integrates a design process and design thinking, focusing on solving complex tasks and repeatedly generating solutions to unknown problems [11]. The education method of DBL involves integrating knowledge from science, mathematics, and engineering into design tasks and producing solutions for systems and objects [28]. Stokholm [29] pointed out that the learning process of DBL is a systematic dynamic thinking mode, including the repetitive process of alignment, research, mission, vision, concept, and product. In the DBL teaching field in Europe and the United States, teachers from the design and engineering fields jointly supervise student teams in carrying out projects. During this process, they also pay attention to teamwork, project planning, and management. The project results include (1) a finished design product report on the design proposal and (2) a design process report on the design process and reflection.

The British Design Council proposed a double-diamond model for the design process [30], including four stages, namely, discovery, definition, development, and delivery (two of which involve divergent thinking, and two of which involve convergent thinking), which is also known as the 4D model. The purpose of DBL is to help high school students learn science and design skills [31]. The course is based on "design", which is used to guide and encourage students to gain scientific knowledge. In addition, through "design", students are exposed to and engaged in practical engineering design methods [32] to improve their abilities in scientific reasoning, self-orientation, and teamwork. Many studies have found that applying the DBL approach to sustainability training in engineering education helps improve students' ability to cope with the practical challenges of sustainability [33]. Students in the DBL group were more outstanding in terms of personal abilities (such as systems thinking, multidisciplinary applications, and collaboration) [34]. The DBL approach has the potential to increase students' desire to learn, to increase their success rates in science classes, and to increase their interest in science topics [33].

This study thus adopts DBL to assist students' learning in a multidisciplinary integrated project course. Teachers provided guidance and tool support in a project course on designing a wearable pet device using Arduino programming, leading students to interact and discuss topics to explore innovative design possibilities. During the learning process, the group of students was encouraged to carry out creative thinking, hands-on practice, concept convergence, and data integration in the design process to accumulate comprehensive and innovative abilities, and thus to solve the problems and to complete the tasks associated with the project together.

## 3. Research Design and Implementation

*3.1. Research Samples*

This study took two classes of students from a junior project course in a technical high school in Southern Taiwan as the experimental objects. One class was an experimental group (42 students) and was subjected to a project course in which DBL programming and the physical computing strategy of Arduino were employed. The other class was a control group (43 students) and was subjected to a project course in which DBL programming was employed alone. The theme was the design of a wearable pet device.

*3.2. Research Design and Process*

This study includes a mix of qualitative and quantitative analysis and adopts the experimental "unequal group pretest and posttest design". As shown in Figure 1, the

18-week project course includes 8 weeks of programming (basic concepts, process control, and sub-programming) teaching, 3 weeks of preliminary exploration on the Internet of Things (IoT) (the sensor principle, IoT communication protocol, Arduino, and Android programming tools introduction), and 7 weeks of project creation (including pretest and posttest). The research design is shown in Table 1.

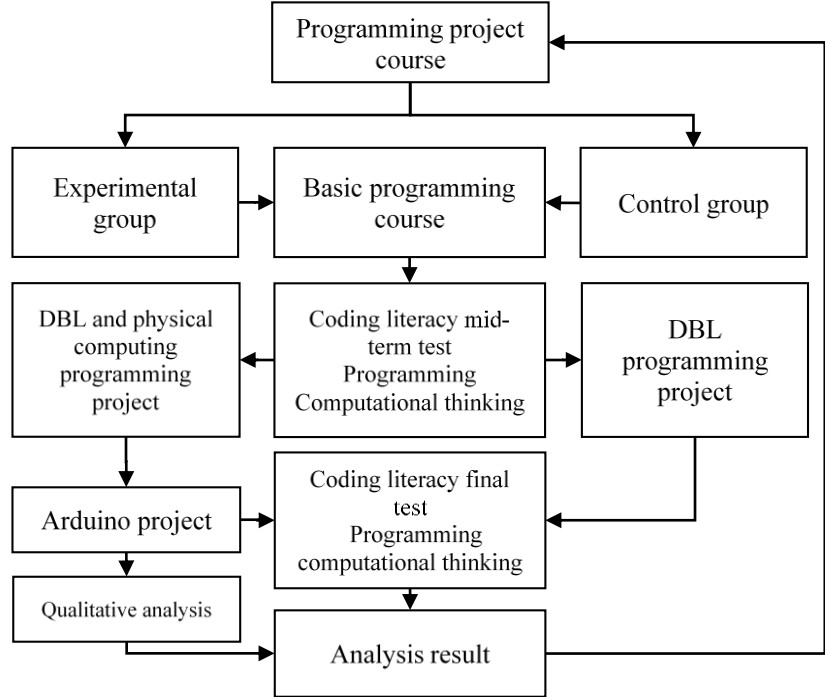

**Figure 1.** Research process.

**Table 1.** Research design.

| O1 | "programming learning effectiveness test and coding literacy" pretest of the experimental group and the control group. |
| --- | --- |
| O2, O3 | "programming learning effectiveness test and coding literacy" posttest of the experimental group and the control group, respectively. |
| X1 | the implementation of the basic programming course. |
| X2, X3 | the implementation of a DBL programming and Arduino project course and a DBL programming project course, respectively. |

The students of the experimental group and the control group followed the same basic programming course (X1) in the first 8 weeks (2 classes per week), and the basic programming course was followed by a programming learning effectiveness test (O1). From the 9th week to the 12th week, the control group was subjected to the DBL programming project course (X3), and the experimental group was subjected to the DBL programming and Arduino project course (X2). Immediately after the end of the experiment, a posttest (O2 and O3) was conducted for the two groups to determine the impact of integrating Arduino into the technical high school project course on the coding literacy of students' programming. At the end of the experimental course, every week, the students in the experimental group filled out a 20-min learning sheet and attended a semi-structured interview, expressing their views on the programming of the project theme and their learning status in the project course and programming. This formed the basis for the qualitative analysis of teaching strategies.

*3.3. Research Tools*

3.3.1. Learning Effectiveness Analysis Tool

This study adopted a self-compiled "coding literacy scale" to examine the effectiveness of this DBL programming and Arduino project course, wherein the "coding literacy scale" includes two dimensions: "programming abilities" and "computational thinking abilities". There are 30 questions on "programming abilities" (Table 2), covering 5 categories: simple calculation, simple judgment, multi-condition judgment, repetitive structure, and comprehensive applications, with 6 questions in each category. There were 103 pre-test students from a technical high school.

**Table 2.** Test questions for programming abilities (example).

| | |
|---|---|
| 1 | What is the function of "if...then...else"? (1) input/output control (2) arithmetic operation processing (3) condition judgment (4) cycle control |
| 2 | Which serial port provided by Arduino provides synchronous or asynchronous serial transmission? I2C (2) SPI (3) ICSP (4) USAR (5) all the above |
| 3 | What is the meaning of delay (200)? (1) generate 200V voltage (2) time delay of 200 s (3) time delay of 200 microseconds (4) time delay of 200 milliseconds (5) all the above |
| 4 | What are the correct identifiers? (1) -a1 (2) a[i] (3) a2_i (4) int t |
| 5 | If "int $n$; float f = 13.8;", then what is the value of $n$ after "$n$ = (int)f%3" is carried out? (1) 1 (2) 4 (3) 4.333333 (4) 4.6 |
| 6 | Which of the following statements can initialize the one-dimensional array a? (1) int a [5] = (0,1,2,3,4,) (2) inta(5) = {} (3) int a[3] = {0,1,2} (4) int a{5} = {10*1} |
| 7 | What is the implied storage class of a variable without a specified storage class? (1) auto (2) static (3) extern (4) register |
| 8 | Suppose there is the following definition: struck sk { int a;float b;}data;int *$p$; What is the correct assignment statement to make $p$ point to a field in the data? (1) $p$ = &a; (2) $p$ = datA, a; (3) $p$ = &datA, a; (4) *$p$ = datA, a |
| 9 | If int a [10] = {1,2,3,4,5,6,7,8,9,10},*$p$ = a, then what is the expression when the value is 9? (1) *$p$ + 9 (2) *($p$ + 8) (3) *$p$ + =9 (4) $p$ + 8 |
| 10 | Which of the following data is a "string constant"? ① 「 a 」 (2) {ABC} (3) 『 abc\0 』 (4) 『 a 』 |

In terms of the test difficulty, the identification degree of Chase [35] was set to 0.2 or above; the difficulty was set to 0.4–0.8 as the selection criteria; and the number of questions in the formal scale was set to 30 after deleting inadequate questions, with a KR20 of 0.86. For "computational thinking abilities", the study referred to the definition of the operational type proposed by ISTE and CSTA [36]. Computational thinking is a problem-solving process, including abstraction, problem decomposition, algorithm thinking, pattern recognition, debugging, and other steps (5 questions). The Likert five-point scale was implemented, with 1–5 points representing "strongly disagree", "disagree", "slightly disagree", "agree", and "strongly agree", respectively. The higher the score, the higher the degree of subjects' agreement with a specific aspect.

3.3.2. Related Software and Hardware Tools

The basic programming course takes C++ language as the textbook content, which includes the fundamental concept, process control, and subprogram of the C++ programming language. In terms of the DBL programming course, this research plans the design with the basic concepts of programming, taking problem solving as the core direction for making wearable pet devices and allowing students to solve problems in groups under the guidance of textbooks (Table 3). For course planning, ESP8266 Wi-Fi MCU was used in this study as the core to construct the practical and theoretical materials, including the Internet of Things (IoT) technology and theory, ESP8266 WiFi MCU programming, sensor

integration design, etc., which allowed students to understand the application and design of Arduino in various fields.

**Table 3.** DBL (design-based learning) Arduino special course design.

| Week | Subject | DBL Arduino Programming Course Content |
|---|---|---|
| 1–2 | Unit 1: A preliminary study on program design and pet wearable device | 1. Pretest questionnaire<br>2. Introduction of programming<br>3. Concept establishment of project development<br>4. Literature collection and reading<br>5. Grouping plan<br>6. Pet wearable devices |
| 3–5 | Application of microcontroller program | 1. C++ language architecture<br>2. Arduino architecture<br>3. Resource utilization of Arduino online platform<br>4. Program conception and reorganization<br>5. Programming teaching |
| 6–8 | LED and sensing switches | 1. LED application<br>2. Light sensing, tilt sensing<br>3. Programming practice<br>4. Design thinking teaching |
| 9–11 | LED and temperature and humidity sensing | 1. Temperature and humidity module<br>2. WIFI transmission<br>3. Programming practice (Description—Execution—Reflection—Debugging—Description)<br>4. Design thinking practice (Discover—Define—Develop—Deliver) |
| 12–16 | Pet wearable device project | 1. Accessory design of pet wearable devices<br>2. Circuit practice<br>3. Programming utility (Description—Execution—Reflection—Debugging—Description)<br>4. Design thinking application (Discover—Define—Develop—Deliver) |
| 17–18 | Project presentation | 1. Project practice report writing<br>2. Design concept exchange<br>3. Publication and display of achievements<br>4. Posttest questionnaire |

### 3.3.3. Text Data

To collect text data on students in their learning process, this study set up a LINE course group (the whole class) and two small groups before the course implementation. The course group is designed for teachers' course announcements, the sharing of learning resources, communication between teachers and students, classroom discussion and the sharing of study sheets, homework feedback, etc. The function of the two small groups includes group discussion, resource sharing, group work feedback, and so on. During the course implementation, students were encouraged to use group discussions and to interact with one another to effectively record and collect feedback sheets and homework experience in each unit. After the course implementation, the student groups were interviewed for 20 min to collect their feelings about learning after completing the DBL Arduino course, which served as the basis for the qualitative analysis.

## 4. Results and Discussion

This study adopted the DBL teaching mode to implement an 18-week Arduino programming project course, conducted experimental teaching to collect and observe the text data from the course and the performance of students in their learning process, and applied

qualitative analysis and induction. Furthermore, a quantitative analysis of "coding literacy" was carried out, supplemented by verification of the effectiveness of the DBL programming Arduino project course, which will be explained in the following subsections.

### 4.1. Performance and Feedback of Students Participating in the DBL Arduino Project

This study employed the four stages of the design process of the Design Council [30], namely, discovery, definition, development, and delivery, and the initial preparation stage of designing the Arduino programming project course in order to guide students to complete the project works. This is explained in this section, which is split into subsections on the basis of the four-stage process.

#### 4.1.1. Preparation Stage—Designing the Theme to Arouse Students' Interest in Learning and to Enhance Their Motivation to Learn Coding

This study claims that the intensity of students' interest in the theme of an activity will affect their participation in subsequent activities. Therefore, our design adopted the theme of "wearable pet device design" to encourage students to integrate what they learned in class and to apply it to the solution of the problems that pets raise at home. According to the survey, the proportion of students who keep pets at home is as high as 69.1%, which is similar to the MIC (Market Intelligence & Consulting Institute) survey results [37], and approximately 66.2% netizens in Taiwan have pet raising experience, with up to 75% of 18–20-year-olds showing a willingness to raise pets. Therefore, most students have a basic understanding of keeping pets, and the study groups were therefore able to quickly focus on the key learning points.

The 8-week basic programming course was implemented, and then, the DBL Arduino project course was taught. Teachers conducted a 3-week Arduino application, teaching IoT, as shown in Figure 2. In the courses, "Arduino Digital and Analog Sensor Implementation" and "Temperature and Humidity Sensor and LCD Display", the students were asked to record sounds using a photosensitive resistor to realize the difference between analog and digital reading and by combining this with a loudspeaker-digital recording output. Sound prompts could be issued when changes in ambient light were sensed. Students often asked questions about the practical application of different sensors in wearable devices, which reflects their learning motivation and problem-solving enthusiasm. Examples include how to confirm the correctness of the Arduino code writing, how to check the correctness of program writing through the free online learning platform software simulation provided by Arduino, and how to access an ESP32 Wi-Fi connection and websites to acquire data to obtain the temperature and humidity of ESP32 +DHT11.

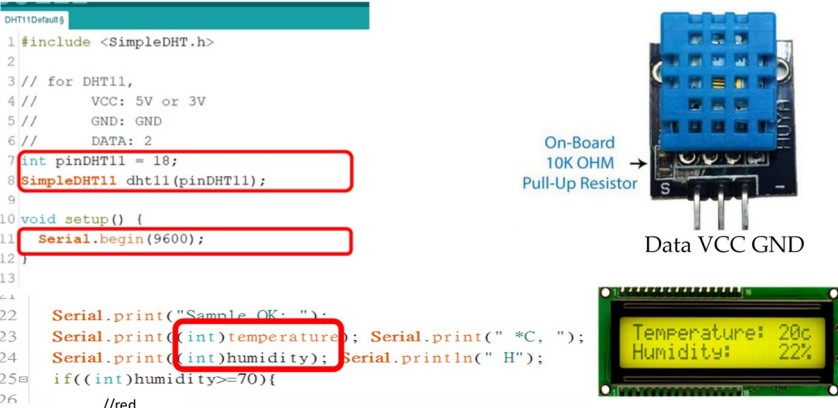

**Figure 2.** Arduino program-controlled temperature and humidity sensing and LCD synchronous display.

| | |
|---|---|
| 04S3 | *To log in to the simulation platform, you need to log in confidentially through your Gmail account and see the verification code on your phone.* |
| 06S2 | *Through the simulation function of the simulation platform, we verify the effect of modifying the speed of an LED flicker.* |
| 05S3 | *The teacher said that the server would be unstable or down on the school network. It is suggested to edit the program code with a notepad or Microsoft Word, save it at any time, and then copy it into the platform simulation program window.* |
| 06S3 | *After we finished the project, we found that we needed to improve our English skills, because the instructions were often misspelled.* |
| 13S2 | *Arduino is really a magical thing! Any function that you may want can be designed by yourself according to what you learn in school. Almost all the knowledge about Arduino is in English, and so I want to study English well.* |

### 4.1.2. Discovery Stage—The Actual Needs of Pets Are Observed in-Depth, and Students' Coding Ability Is Strengthened

Through various discussions, the students decided on the best pet observation tool. The teacher guided the students to observe the interaction between pets and owners in an empathetic way to make new discoveries and to uncover people's deeper needs. Most groups chose dogs as the most popular pet. The reason for this is that many families have raised dogs, and they understand their habits in daily life. Thus, as a functional plan of wearable pet devices, the student groups were asked to draw preliminary design sketches of the wearable devices discussed, as shown in Figure 3.

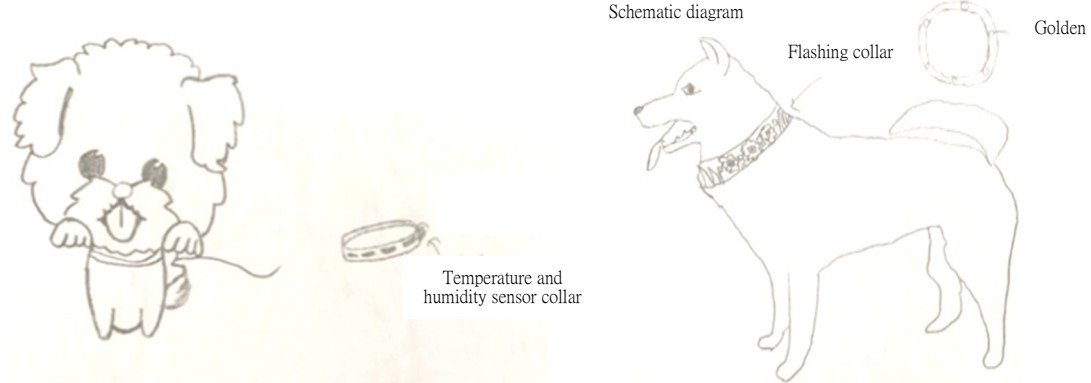

**Figure 3.** Design drawings of the students' works.

| | |
|---|---|
| 13S1 | *The pet object is a mixed Shih Tzu raised by my family.* |
| 16S1 | *Our group worked out a way to figure out who would be responsible for providing the dog for our experiment.* |
| 15S2 | *We discussed the types of pets (large, medium, and small dogs) and experimental devices that could be worn by dogs. Finally, we chose the middle-sized dog of the group leader's family as the experimental subject.* |

In terms of the application of Arduino programming, an open and independent learning environment was created to inspire students' imagination through group cooperation and discussion. Thus, they were asked to understand obscure concepts of science and program coding. For example, students hoped to apply the sensing principle of photosensitive resistance to wearable devices. However, most students easily made mistakes in the relationship between light and resistance. The teacher helped the students learn how to change the light level of the photosensitive resistor through an actual experiment. In this way, students could successfully simulate the Arduino learning platform and to complete an LED light bar function test, as shown in Figure 4.

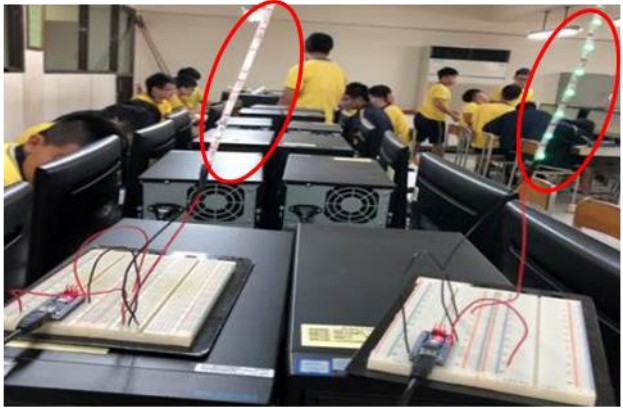

**Figure 4.** Students using the Arduino learning platform to simulate and test an LED light bar.

*Student comments*

| | |
|---|---|
| 03S1 | *The effect of covering the photosensitive resistor by hand was not good. The teacher said that the change of light is based on the unit of "lumens". It was suggested that we search for information online again.* |
| 08S3 | *I turned on the flashlight of the mobile phone, but it did not respond. This is related to the fact that the photosensitive resistor is a negative resistance component. It is probably too bright and needs to be shaded.* |

### 4.1.3. Definition Stage—Demands Are Integrated to Identify Practical Problems, and Students' Confidence in Applying Coding to Solve Problems Is Enhanced

The teacher instructed the students to integrate and summarize all the possible demands observed in the discovery stage and to try to connect all the demands. For example, increasing the safety of pet dogs during the night, monitoring the temperature and humidity of pet dogs, and anti-loss devices, etc., are real and definite problems. Based on this, more specific functional design drawings of wearable devices were drawn, as shown in Figure 5.

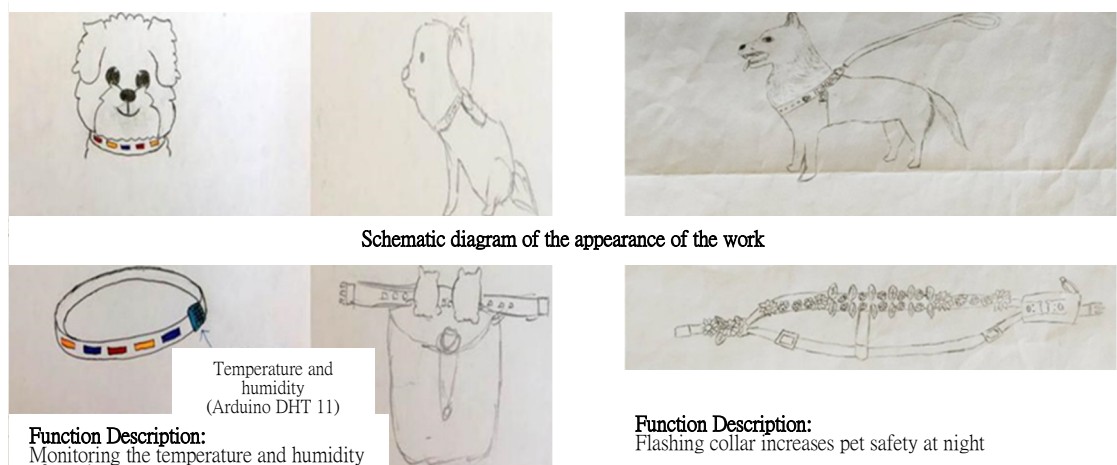

**Figure 5.** Functional design drawings of the students' works.

*Student comments*

| | |
|---|---|
| 11S3 | *The Arduino board and the three primary colors of light (red R, green G, and blue B) were used to make the LED light bar show different colors, representing changes of temperature, causing more creativity and ideas to be applied in daily life.* |
| 11S1 | *The RGB LED light bar has a low power consumption, rich color, and brighter light than ordinary bulbs. It is practical and suitable for wearable pet devices.* |
| 13S2 | *The RGB LED light bar can be adjusted to a wide variety of colors through the program code design. When I visited the lighting fair, I found that many beautiful lights came from RGB LEDs.* |

In terms of the application of Arduino programming, the students were asked to apply the function of modular parts of the Arduino temperature and humidity sensors according to functional requirements. Therefore, students set about writing the function library program and the user interface settings with sensor changes, with the aim of achieving a data transmission conversion display. Therefore, the students learned that temperature and humidity sensing is based on the scientific knowledge of analog-to-digital conversion. Through the input signals of temperature and humidity, the module performs data processing and outputting. Therefore, the sensor signal serves as the source of the element's trigger signal.

*Student comments*

| | |
|---|---|
| 06S1 | *You can breathe through the mouth as a way of simulating the network platform, and you can see the temperature change clearly through the generated numerical values.* |
| 06S2 | *After the program is written, you can use cold air to blow the components and then use the sunlight for illumination, and you can see the change of temperature.* |
| 06S3 | *The correctness of the program writing and temperature sensing functions can be tested by using a hair dryer to raise the temperature.* |

### 4.1.4. Development Stage—Tests Are Repeated, and Solutions Are Verified to Strengthen Students' Code Debugging Abilities

According to the problems identified in the definition stage, a variety of improvement strategies were carried out, and the student teams were encouraged to put forward improvement plans and to repeat experiments, such as creation, prototyping, functional testing, and iterative solutions. A hands-on practical activity demonstrated the students' ability to integrate theoretical disciplines and the application and debugging of code, thus improving and perfecting the innovative ideas of the student team.

*Student comments*

| | |
|---|---|
| 03S1 | *The project uses Arduino to control LED lighting, which is a very new experience. When I first experienced Arduino, I often encountered many obstacles in the process of the project, such as circuit errors, light bar failure, and program errors, which we had to find ways to overcome.* |
| 05S1 | *If you encounter problems or have needs in life, then you can try to use the function of an Arduino logic single-chip microcontroller, combined with the principle and programming of electronic circuits, to complete a functional test using the wiring of a breadboard tester.* |

In terms of the application of Arduino programming, this project combined the software program with a hardware circuit and paid attention to the function of work presentation. Therefore, the team members were asked to complete the program writing and circuit made jointly through a division of labor. During the process, program simulation software, welding tools, and instrument testing were used.

*Student comments*

| | |
|---|---|
| 05S1 | *When testing the function of the LED lamp and finding that the LED lamp was not on, we instituted a division of labor to find out whether there was any error in the circuit or program code.* |
| 03S3 | *The problem of having an incorrect code input often occurred during the testing of the circuit. Through inspections conducted by the team members, we uncovered the error and successfully corrected the program code.* |
| 13S3 | *During the functional testing of the wearable device, the dog would try to break free of the collar and destroy the RGB light bar. Therefore, we put the RGB light bar into a transparent water pipe for protection and overcame the difficulty.* |

### 4.1.5. Delivery Stage—Group Cooperation to Produce Physical Works and Optimize Students' Coding Application Abilities

This project required each student team to complete physical work and to present and communicate the results. Therefore, in addition to confirming that the functionality of the display of the wearable device was correct, the student teams needed to give the work an aesthetic design that is consistent with the overall style. Through a process of group cooperative learning, students with good information abilities help other students improve their coding abilities and optimize their information technology literacy and hands-on abilities.

*Student comments*

| | |
|---|---|
| 06S2 | *The color of the RGB LED light bar must be adjusted to pink to match Niu-Niu's collar.* |
| 04S1 | *After understanding the basic principle and practical operation of Arduino, we can refer to the sample works of the website and then carry out circuit and programming according to our own needs, so as to complete the works within the time limit.* |

In terms of Arduino and programming applications, students need to fully understand the characteristics and principles of wearable device parts. On the basis of their acquired knowledge, students could find problems, simulate hypotheses, formulate solutions, write programs, make circuits, perform functional tests, and then complete the works, as shown in Figure 6a,b. In this way, it was helpful to improve students' integrated application of interdisciplinary knowledge and abilities, such as circuit board welding, wiring, program-controlled design, and other circuit function control; measurement, computing, and other mathematical practices, from design to production; the application of electrical principles, such as batteries, wires, and currents; and the application of scientific knowledge in components, material selection, tool use, etc. Finally, in combination with the project, the finished structures were designed, and the works, with unique group images, were produced.

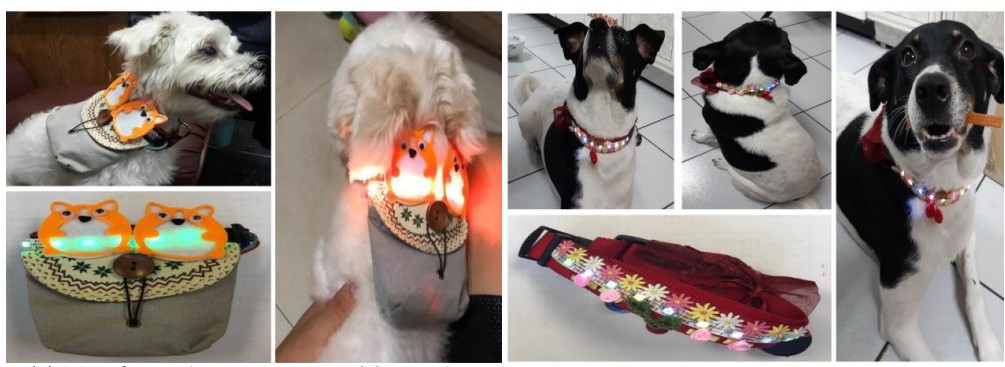

**(a) Mood-Sensing RGB Wearable Device**          **(b) Safe LED Wearable Device**

**Figure 6.** Students' works.

| *Student comments* | |
|---|---|
| 02S1 | *We designed a "mood sensing RGB wearable device", which integrates the functions of a microcontroller and tilt sensor and uses a "ball switch" to make two iron beads on the interior form a guiding mode to generate signals by shaking, so as to detect the movements and responses of pets.* |
| 02S2 | *When the pet is happy, it will generate signals in response to jumping, running, and spinning and thus trigger the RGB movement, so that the owner can understand the pet's mood at that time. When the pet is in a low mood or uncomfortable, it will move slowly and have no vitality.* |
| 02S3 | *In terms of the aesthetic design, we designed a cute pattern decoration and stored the circuit in the bag to protect the LED and improve the safety of the circuit.* |
| 02S4 | *We added the design of a small backpack, which can carry the items needed for walking the dog, which is easy to carry.* |
| 03S1 | *In order to increase the safety of walking pet dogs at night, photosensitive resistance characteristics and program writing were used to design a safe wearable LED device that automatically lights up, so as to improve drivers' attention to dogs being walked in the street.* |
| 03S2 | *We stuck the light bars on the outside to improve pet dogs' comfort in wearing the device and designed three lighting modes (constant lighting, intermittent lighting, and fast lighting) to improve the warning signals.* |
| 03S3 | *Our pet is female, and so we used a lace pattern to improve the aesthetic aspects of the finished product.* |
| 03S4 | *An Arduino NANO board was used to replace the complex integrated circuit, and the conductor was hidden in the collar. After many actual measurements and trial wears, we improved its comfort and aesthetic aspects.* |

### 4.2. Effectiveness Analysis of Students' Coding Literacy

Before and after the course implementation, the research team conducted a questionnaire survey among the students in the experimental group (42 students) and the control group (43 students). In terms of background variables, the average age of the students in the experimental group was 18.28 years old, and that of the students in the control group was 18.34 years old. They were all senior students at a technical high school. Coding literacy in this study includes "programming abilities" and "operational thinking abilities". To prevent the different average class performances interfering with the experimental results, Analysis of Covariance (ANCOVA) was adopted for the questionnaire results, with the pretest scores as the covariable and the posttest scores as the dependent variable, in order to examine the influence of the DBL Arduino course on the experimental student group's programming abilities and operational thinking abilities, which are described as follows.

#### 4.2.1. Analysis of Students' Programming Abilities

This study examined the students' programming abilities. Five types of questions were planned, including simple calculation, simple judgment, multi-condition judgment, repetitive structure, and comprehensive applications. The analysis results are shown in Table 4.

The "Levene test equation of error variance" was first examined, and the F values were 0.166 ($p = 0.684$), 0.059 ($p = 0.808$), 0.514 ($p = 0.475$), 0.325 ($p = 0.570$), and 2.067 ($p = 0.153$). The significance level was greater than 0.05, indicating that there was no significant difference in the error variation of the two groups according to the variables, and the two groups were thus homogeneous. Therefore, covariate analysis could be performed. After reviewing the "test of the inter-subject effect items", it was found that the t values of "simple calculation" and "simple judgment" were −1.381 ($p = 0.169$) and −1.830 ($p = 0.069$), respectively, and that there was no significant difference. The t values of "multi-condition judgment", "repetitive structure", and "comprehensive applications"

were $-2.037$ ($p = 0.043$), $-2.039$ ($p = 0.043$), and $-2.056$ ($p = 0.042$), respectively, and the significance of each of these values was less than 0.05, thus achieving a significant difference. The adjusted average of the experimental group's "multi-condition judgment" was 4.36, which is higher than the control group's 4.10. The adjusted average of "repetitive structure" was 4.48, which is higher than the control group's 4.23, and the adjusted average of "comprehensive applications" was 4.21, which is higher than the control group's 3.92.

**Table 4.** Analysis of covariance (ANCOVA) summary of students' programming abilities.

| Item | Mean | Standard Deviation | Adjusted Mean | Adjusted Standard Deviation | Effect Item Test | |
|---|---|---|---|---|---|---|
| | | | | | *t* | *p* |
| Simple calculation | 4.09 | 0.804 | 4.11 | 0.081 | −1.381 | 0.169 |
| | 4.33 | 0.758 | 4.30 | 0.109 | | |
| Simple judgment | 3.94 | 0.931 | 3.97 | 0.094 | −1.830 | 0.069 |
| | 4.32 | 0.893 | 4.26 | 0.126 | | |
| Multi-condition judgment | 4.06 | 0.773 | 4.10 | 0.076 | −2.037 | 0.043 |
| | 4.43 | 0.693 | 4.36 | 0.102 | | |
| Repetitive structure | 4.21 | 0.731 | 4.23 | 0.071 | −2.039 | 0.043 |
| | 4.52 | 0.638 | 4.48 | 0.095 | | |
| Comprehensive application | 3.88 | 0.878 | 3.92 | 0.083 | −2.056 | 0.042 |
| | 4.28 | 0.743 | 4.21 | 0.111 | | |

The results showed that the experimental student group exhibited a significantly better programming performance in "multi-condition judgment", "repetitive structure", and "comprehensive applications" in the DBL Arduino project course than the control group did. This is consistent with the results of García-Peñalvo, Reimann, and Maday [38]. However, there was no significant difference between the two groups in "simple calculation" and "simple judgment".

### 4.2.2. Analysis of Students' Computational Thinking Abilities

This study examined the students' computational thinking abilities. Five types of questions were asked, including abstraction, problem decomposition, algorithm thinking, pattern recognition, and debugging. The analysis results are shown in Table 5.

The "Levene test equation of error variance" was first examined, and the F values were 0.030 ($p = 0.862$), 1.369 ($p = 0.244$), 1.150 ($p = 0.258$), 0.539 ($p = 0.464$), and 1.710 ($p = 0.357$). The significance level was greater than 0.05, indicating that there was no significant difference in the error variation of the two groups according to the variables, and the two groups were thus homogeneous. Therefore, covariate analysis could be performed. After reviewing the "test of the inter-subject effect items", it was found that the t values of "abstraction" and "problem decomposition" were $-0.743$ ($p = 0.459$) and $-0.910$ ($p = 0.364$), respectively, and that there was no significant difference. The t values of "algorithm thinking", "pattern recognition", and "debugging" were $-2.121$ ($p = 0.035$), $-2.016$ ($p = 0.046$), and $-0.211$ ($p = 0.036$), respectively, and the significance of each of the values was less than 0.05, indicating a significant difference. The adjusted average of the experimental group's "algorithm thinking" was 4.67, which is higher than the control group's 4.43. The adjusted average of "pattern recognition" was 4.54, which is higher than the control group's 4.32, and the adjusted average of "debugging" was 4.57, which is higher than the control group's 4.29.

**Table 5.** ANCOVA summary of students' computational thinking abilities.

| Item | Mean | Standard Deviation | Adjusted Mean | Adjusted Standard Deviation | Effect Item Test | |
|---|---|---|---|---|---|---|
| | | | | | $t$ | $p$ |
| Abstraction: when I need to record data, I will think of a "list" | 4.31 | 0.651 | 4.36 | 0.066 | −0.743 | 0.459 |
| | 4.52 | 0.632 | 4.44 | 0.089 | | |
| Problem decomposition: I will try to decompose the problem into several small problems to solve | 4.42 | 0.663 | 4.46 | 0.066 | −0.910 | 0.364 |
| | 4.62 | 0.595 | 4.56 | 0.089 | | |
| Algorithm thinking: when I encounter problems, I will try to think out the steps to solve them | 4.42 | 0.679 | 4.43 | 0.065 | −2.121 | 0.035 |
| | 4.69 | 0.503 | 4.67 | 0.087 | | |
| Pattern recognition: I can see the similarities and differences in each problem | 4.29 | 0.636 | 4.32 | 0.063 | −2.016 | 0.046 |
| | 4.58 | 0.569 | 4.54 | 0.085 | | |
| Debugging: when the program is wrong, I will try to find the problem and fix it | 4.26 | 0.750 | 4.29 | 0.069 | −0.211 | 0.036 |
| | 4.60 | 0.531 | 4.57 | 0.093 | | |

The results showed that the experimental student group exhibited a significantly better computational thinking performance in "algorithm thinking", "pattern recognition", and "debugging" in the DBL Arduino project course than the control group. This is consistent with the results of Lu, Hong, Chen, and Ma [7]. However, there was no significant difference between the two groups in "abstraction" and "problem decomposition".

## 5. Global Discussion

The qualitative and quantitative analysis results are discussed in this section.

### 5.1. Students' Advanced Programming Abilities Were Significantly Improved through the Phased Teaching Content Design

This DBL Arduino project course adopted a phased teaching content design, including a "Basic Programming Course" (8 weeks) and " Arduino Advanced Course" (3 weeks). The basic course aimed to strengthen the students' programming abilities, considering the differences between students in terms of their programming abilities, using several examples, and designing different levels of questions to understand their different levels of learning effectiveness. The advanced course focused on applying Arduino, providing students with opportunities to try and solve problems. The majority of students acquired programming abilities step-by-step through learning the above phased teaching content [38]. The experimental student group performed significantly better than the control group in terms of

advanced programming abilities in multi-condition judgment, repetitive structure, and comprehensive applications.

### 5.2. Students' Computational Thinking and Practical Application Abilities Were Significantly Improved through the Practical Teaching Activity Design

This DBL Arduino project course planned Arduino practical teaching activities with a high practicality. During the course, most students were positive about incorporating novel, operational, and real-time responsive hardware devices into the programming course, which showed that Arduino open hardware could have a potential motivational influence on the application of programming teaching. In practical teaching activities, most students try to decompose problems and to think about solutions and steps when encountering problems [10]. The experimental student group performed significantly better than the control group in terms of computational thinking and practical application abilities in algorithm thinking, pattern recognition, and debugging.

### 5.3. Authentic Theme Planning and Problem Solving Can Help Students Improve Their Coding Literacy

This DBL Arduino project course involved the design of an authentic wearable pet device and received mostly positive student feedback, showing that the students were interested in the theme and that they were effectively motivated to learn coding, which was an important part of the preparation stage of this course. Moreover, through the guidance of the four stages of DBL (discovery, definition, development, and delivery), the students learned how to control Arduino open hardware through the associated program to solve the problems and to satisfy the requirements of the interaction between pets and owners. In the process, students' application abilities and practical experience improved [39]. Furthermore, the process of discovering and solving problems repeatedly enhanced students' confidence and debugging abilities in applying coding to solve problems. It also stimulated them to develop innovative ideas concerning the application of science and technology and thus improved their abilities in applying coding [7,40].

## 6. Conclusions and Suggestions

In an era of vigorous information technology development, a trend has emerged of integrating information technology into everyday life. Therefore, there is a global consensus that coding literacy, which is related to computer science, logical thinking, computational thinking, etc., needs to be cultivated in students. In view of this, this study adopted the DBL teaching strategy, integrated the physical computing strategy of Arduino into the programming course, and employed an interesting project involving the design of a wearable pet device to provide students with an environment for the practical application of programming and opportunities to experience technology skills and information processing, with the aim of inspiring them to explore new ideas and perspectives. Moreover, the four stages of DBL were used to guide students through the repeated process of discovering and solving problems to enrich and improve the communication of programming and technology application knowledge among students and to further improve students' coding literacy and computational thinking abilities. Therefore, the project course, "wearable pet device design using DBL and Arduino", employed in this study can provide a reference for the implementation of programming courses in technical high schools.

While the results of this study show that teaching students to apply programming knowledge to control Arduino-related sensors through hands-on activities can motivate them to engage in programming courses, it also offers a good learning effectiveness. Because the subjects of this research were technical high school students with basic electronic literacy, the results herein should not be applied to students from different departments. Since Arduinos are less accessible to most students in nontechnical high schools and electronic-related departments, teachers need to pay attention to the problem of how to reduce the difficulties in teaching or learning associated with the hardware itself in course

implementation. In this study, a LINE course group was established to provide a messaging platform for teacher–student interaction and peer communication, and relevant learning materials, course examples, and teaching videos were uploaded to the message group notepad for students' reference and further study.

**Author Contributions:** Conceptualization: S.-J.L.; methodology: C.-C.C.; validation: S.-J.L.ou and C.-C.C.; formal analysis: C.-C.C.; resources: C.-C.C.; writing—original draft preparation: C.-C.C.; writing—review and editing: S.-J.L.; supervision: S.-J.L.; funding acquisition C.-C.C. All authors have read and agreed to the published version of the manuscript.

**Funding:** This research was funded by the Ministry of Science and Technology (MOST) grant number MOST 108-2511-H-020 -002 -MY3.

**Institutional Review Board Statement:** Not applicable.

**Informed Consent Statement:** Informed consent was obtained from all subjects involved in the study.

**Data Availability Statement:** MDPI Research Data Policies.

**Conflicts of Interest:** The authors declare no conflict of interest.

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
