# Peer review of "Physical Computing Strategy to Support Students’ Coding Literacy: An Educational Experiment with Arduino Boards"

_applsci, doi:10.3390/app11041830_

Round 1
Reviewer 1 Report
This paper describes a trial of using a design-based learning approach to teach Arduino coding to a group of high-school students in Taiwan. I found the paper very readable and thorough in its background and literature review. I think this study is worth publishing.
The authors were clear in writing the background, reason for the study, and what the students actually experienced and did. The analysis of how well the students performed was less clear to me and this section (4.2) could be worked on to make it clearer. I did find it difficult to follow the authors' statistical analysis of student learning. Who took the measurements of student learning on which the statistics were based? How were the students assessed by the teachers? How old were the students, on average? Do the results from this study have any lessons for other high-school educators teaching programming?
A few minor points:
I spotted several typographical errors here and there.
The student comments may better be arranged into tables.
Line 164, page 4: a reference to figure 1 appears to go nowhere.
Section 3, line 180. This sentence is a leftover from the document template.
Figure 4: The authors might consider making the right-side photo a close-up of the components on the breadboard.
Figure 6 may not be necessary.
Author Response
Response to Reviewer 1 Comments
Point 1: (x) Moderate English changes required. 

Response 1:
English-Editing-Certificate-26872
Point 2: This paper describes a trial of using a design-based learning approach to teach Arduino coding to a group of high-school students in Taiwan. I found the paper very readable and thorough in its background and literature review. I think this study is worth publishing.
Response 2: Thank you.
Point 3: The authors were clear in writing the background, reason for the study, and what the students actually experienced and did. The analysis of how well the students performed was less clear to me and this section (4.2) could be worked on to make it clearer. I did find it difficult to follow the authors' statistical analysis of student learning. Who took the measurements of student learning on which the statistics were based? How were the students assessed by the teachers? How old were the students, on average? Do the results from this study have any lessons for other high-school educators teaching programming?
Response 3: Thank you. We had added the detail descriptions, as following, for the analysis of students' learning effect. (in red)
Who took the measurements of student learning on which the statistics were based? How were the students assessed by the teachers? How old were the students, on average?
Section 3.3.1 Learning Effectiveness Analysis Tool
Table 2. Test questions for programming abilities (Example). (page, 6, Section 3.3.1, line 240, 254-255)
Section 3.3.2 Related Software and Hardware Tools
Table 3. DBL Arduino special course design. (page, 7, Section 3.3.2, line 262, 267-268)
Section 3.3.3 Text Data (page, 8, Section 3.3.3, line 270)
Section 4.2 Effectiveness Analysis of Students' Coding Literacy (page, 13, Section 4.2, line 425-437)
Do the results from this study have any lessons for other high-school educators teaching programming?
Section 5. Global Discussion (page, 15, Section 5, line 491-531)
A few minor points:
I spotted several typographical errors here and there.
The student comments may better be arranged into tables.
We had added the table for each of students’ comments. (Section 4.1, line 321, 355, 348, 363, 374, 389, 396, 407, 421)
Line 164, page 4: a reference to figure 1 appears to go nowhere.
We had deleted the figure1.
Section 3, line 180. This sentence is a leftover from the document template.
We had revised it. (line 190.)
Figure 4: The authors might consider making the right-side photo a close-up of the components on the breadboard.
We had revised the Figure 4. (, Section 4.1.2, line 350)
Figure 6 may not be necessary.
We had deleted Figure 6.

Reviewer 2 Report
Comments on the manuscript "Physical Computing Strategy to Support Students' Coding Literacy: An Educational Experiment on the Arduino Board":
- The abstract should start with a short introduction to the subject. The purpose comes next. It should be focused on the reader.
- What is the objective of a revised literature in the manuscript? Would it be better to call it "Background" or "Framework"?
- It would be interesting to highlight where other works have successfully applied this methodology.
- In lines 235-236: in which lines of ref. 31?
- 3.3.3 subsection should be expanded. The process is unclear.
- Consider updating the bibliography:
https://doi.org/10.25217/ji.v5i1.773
https://doi.org/10.3390/educsci10090225
- If section 4 is called "Results and Discussion", it is not logical to call section 5 "Discussion" again. Try "Global Discussion" or include them in section 4 as a suggestion.
Author Response
Response to Reviewer 2 Comments
Point 1: (x) Moderate English changes required. 

Response 1:
English-Editing-Certificate-26872
Point 2: Comments on the manuscript "Physical Computing Strategy to Support Students' Coding Literacy: An Educational Experiment on the Arduino Board":
- The abstract should start with a short introduction to the subject. The purpose comes next. It should be focused on the reader.
- What is the objective of a revised literature in the manuscript? Would it be better to call it "Background" or "Framework"?
- It would be interesting to highlight where other works have successfully applied this methodology.
- In lines 235-236: in which lines of ref. 31?
- 3.3.3 subsection should be expanded. The process is unclear.
- Consider updating the bibliography:
https://doi.org/10.25217/ji.v5i1.773
https://doi.org/10.3390/educsci10090225
- If section 4 is called "Results and Discussion", it is not logical to call section 5 "Discussion" again. Try "Global Discussion" or include them in section 4 as a suggestion.
Response 2:
We had added the detail descriptions, as following, for the Comments and Suggestions of Reviewer. (in red)
- The abstract should start with a short introduction to the subject. The purpose comes next. It should be focused on the reader.
We had revised the abstract. (line 9-29)
- What is the objective of a revised literature in the manuscript? Would it be better to call it "Background" or "Framework"?
We had revised it. (Section 1 Background, line 32)
- It would be interesting to highlight where other works have successfully applied this methodology.
We had added some literatures to highlight the successfully applied case in this methodology. (Section 2.2. Physical Computing, line 147-150; Section 2.3. Design-based Learning, line 176-182)
- In lines 235-236: in which lines of ref. 31?
We had revised it. (Section 3.3.1, line 248)
ISTE & CSTA. (2011). Operational Definition of Computational Thinking for K-12 Education. Retrieved January 15, 2018, from https://id.iste.org/docs/ct-documents/computational-thinking-operational-definition-flyer.pdf?sfvrsn=2
- 3.3.3 subsection should be expanded. The process is unclear.
We had added detail descriptions to expand the process. (Section 3.3.3, line 270-281)
- Consider updating the bibliography:
https://doi.org/10.25217/ji.v5i1.773
https://doi.org/10.3390/educsci10090225
We had updated the Article citations. ([25-27] line 609-616; [33, 34] line 628-631; [39] line 640-641)
- Chen, Y., Li, W., Zeng, J., Ye, J., & Luo, Y. (2018. Oct. 24-26). Analysis and analysis of the cognitive learning effectiveness of IoT module programming for non-information related students. The 24th Taiwan Academic Network Conference (2018 TANET), Taiwan, Taoyuan. DOI: 10.6861/TANET.201810.0111
- Hsu, M.-J., Ho, C.-P. (2018). Learning Outcomes of College Students' Internet of Things – Taking the Teacher-student Interaction as the mediator. JOURNAL OF CAGST, 105-118.
- Pratiwi, U., Al Haddar, G., & Kristiawan, M. (2020). Arduino-Based Mini Reed Switch Magnetic Sensor Media: Implementation in Physics Learning to Improve Students’ Analyzing Ability. Jurnal Iqra': Kajian Ilmu Pendidikan, 5(1), 183-193. https://doi.org/10.25217/ji.v5i1.773
- Doppelt, Y., Mehalik, M. M., Schunn, C. D., Silk, E., & Krysinski, D. (2008). Engagement and achievements: A case study of design-based learning in a science context. Journal of technology education, 19(2), 22-39.
- Huang, Z., Peng, A., Yang, T., Deng, S., & He, Y. (2020). A design-based learning approach for fostering sustainability competency in engineering education. Sustainability, 12(7), 2958.
- González-Zamar, M. D., & Abad-Segura, E. (2020). Implications of virtual reality in arts education: Research analysis in the context of higher education. Education Sciences, 10(9), 225.
- If section 4 is called "Results and Discussion", it is not logical to call section 5 "Discussion" again. Try "Global Discussion" or include them in section 4 as a suggestion.
We had revised the Section 5 "Global Discussion". (Section 5, line 491)

Reviewer 3 Report
Interesting paper and teaching students about coding is very important.
For this pre-post test design factors affecting the internal and external validity are not clearly presented.
Author Response
Point 1: (x) English language and style are fine/minor spell check required
Response 1:
English-Editing-Certificate-26872
Point 2:
Interesting paper and teaching students about coding is very important.
For this pre-post test design factors affecting the internal and external validity are not clearly presented.
Response 2:
We had added the detail descriptions, as following, for the Comments and Suggestions of Reviewer. (in red)
We had revised the abstract. (line 9-29)
We had added some literatures to highlight the successfully applied case in this methodology. (Section 2.2. Physical Computing, line 147-150; Section 2.3. Design-based Learning, line 176-182)
Section 3.3.1 Learning Effectiveness Analysis Tool
Table 2. Test questions for programming abilities (Example). (page, 6, Section 3.3.1, line 240, 254-255)
Section 3.3.2 Related Software and Hardware Tools
Table 3. DBL Arduino special course design. (page, 7, Section 3.3.2, line 262, 267-268)
We had added detail descriptions to expand the process. (Section 3.3.3, line 270-281)
Section 4.2 Effectiveness Analysis of Students' Coding Literacy (Section 4.2, line 425-437)
Section 5. Global Discussion (Section 5, line 491-531)
We had updated the Article citations. ([25-27] line 609-616; [33, 34] line 628-631; [39] line 640-641)

Reviewer 4 Report
The research aims to explore the impact of the integration of a physical computing strategy into a programming project course on the coding literacy of technical high school students. The DBL (design-based learning) teaching was implemented with two classes of students in a technical high school in southern Taiwan, sampling one class as a control group (43 students) carrying out the DBL programming project and the other class as an experimental carrying out the DBL programming Arduino project of a physical computing strategy. The study also carried out pre-test, post-test, and questionnaire survey to conduct research analysis through the t-test of paired samples.The results obtained indicates that the experimental group students exhibited significantly better computational thinking ability in "algorithm thinking", "pattern recognition", and "de-bugging" after the DBL Arduino project course than the control group.
The paper is well structured. Information is clearly presented. Good and consistent methodology and research tools are used. All phases of the research are clearly described and also illustrated. Results are relevant and can be used for improving the implementation of programming courses in technical high schools.
Author Response
Thank you.

Round 2
Reviewer 2 Report
Dear authors,
The revised manuscript has improved after the last corrections.
This manuscript is a resubmission of an earlier submission. The following is a list of the peer review reports and author responses from that submission.